# Association of gut microbiota with depression post-myocardial infarction: A systematic evaluation and meta-analysis protocol

Xiang Liu[1], Xiaojun Shi[1], Haibin Zhao[2]*, Jiqiu Hou[2], Weizhe Zhao[1], Wanli Ding[1]

1 Beijing University of Chinese Medicine, Beijing, China, 2 Dongfang Hospital of Beijing University of Chinese Medicine, Beijing, China

* Haibin999@126.com

**Data Availability Statement:** No datasets were generated or analysed during the current study. All relevant data from this study will be made available upon study completion.

## Abstract

### Background

Depression post-myocardial infarction (MI) is becoming more prevalent. The gut-brain axis (GBA), influenced by the gut microbiota, is a critical component in understanding depression post-MI. Despite the well-established connection between gut microbiota and depression post-MI, this relationship remains incompletely understood.

### Methods and analysis

This protocol will follow the Preferred Reporting Items for Systematic Review and Meta-analysis Protocol (PRISMA-P) 2020 statement. Beginning from inception to October 2023, a systematic search will be conducted across eight electronic databases, including PubMed, MEDLINE, Scopus, Embase, Cochrane Clinical Trials Database, Web of Science, China National Knowledge Infrastructure, and China Biomedical Literature Database. Pre-selected studies will be independently assessed by two researchers following a standard inclusion, data extraction and quality assessment protocol. The primary outcome measures are differences in the profile of gut microbiota and rating scale scores for depression. Fixed-effects models will be used when both clinical heterogeneity and statistical heterogeneity are low, otherwise random-effects models will be used. Furthermore, subgroup analyses will be conducted on the depression severity of the participants using the same psychiatric scales employed, study type and geographic region. Random forest plot runs and research-related statistical analyses will be carried out using Rev Man V.5.3 software.

### Expected results

This study will identify the association between the gut microbiota and the onset of depression post-MI, and provide evidence for the use of probiotics as an adjunctive treatment for depression post-MI.

### Trial registration

**Prospero registration number:** CRD42023444026.

**Funding:** the National Natural Science Foundation of China (grant number No. 82174332). The funders had no role in study design, data collection and analysis, decision to publish, or preparation of the manuscript.

**Competing interests:** The authors have declared that no competing interests exist.

# 1 Introduction

Depression is expected to emerge as the primary contributor to global somatic disease over the next decade [1]. Cardiovascular disease, currently the leading cause of death on the planet, along with depression, poses significant challenges in terms of long-term disability development [2,3]. A well-established bidirectional relationship exists between cardiovascular disease and psychiatric disorders [4]. Patients with Coronary Artery Disease (CAD) commonly exhibit depressive symptoms such as low mood, decreased interest or pleasure, and low self-esteem. After MI, patients face an elevated risk of depression, with an estimated 15–35% developing major depression [5,6] and around 20% experiencing mild depression or depressive symptoms [7]. Depression in these patients may lead to medication noncompliance and high dropout rates from cardiac rehabilitation programs [8–10]. Major depression is independently associated with an increased risk of morbidity and mortality from CAD [11]. Individuals with a history of depression are four times more likely to experience an acute MI compared to those without a history of depression [12]. This increased risk will affect cardiovascular pathophysiology, including blood viscosity, vasodilatory tone, plasma volume, blood pressure, heart rate, and vascular resistance [13].

A promising area of research is the microbiota-GBA, which may shed light on the bidirectional relationship between cognitive function, mood, and gastrointestinal disorders. The gut microbiota, composed of trillions of bacteria, viruses, archaea, and fungi, maintains a dynamic equilibrium under physiological conditions. The prevailing hypotheses regarding the mechanisms underlying the relationship between gut microbiota and the Central Nervous System (CNS) include dysfunction of 5-hydroxytryptamine (5-HT) dysfunction, altered HPA axis function, exocrine signaling, and inflammatory responses. It is believed to affect cognitive function and mood through neural, metabolic, hormonal, and immune-mediated mechanisms [14–20].

Meta-analysis is a method to synthesize available resources to determine the consistency of studies that show a significant correlation between gut microbiota dysbiosis and depression post-MI. Despite growing evidence, but the exact relationship between those lacks consensus. Therefore, this study aims to present a systematic protocol for assessing the association between gut microbiota and depression post-MI, in order to further investigate the comorbid mechanisms involved.

# 2 Methods

## 2.1 Study design

The meta-analysis protocol has been registered with the PROSPERO Network's International Prospective Systematic Review Registry (registration number: CRD42023444026). The consent for this protocol was obtained based on the guidelines of the Preferred Reporting of Systematic Reviews and Meta-Analysis Protocols Project (PRISMA-P) 2020 statement [21].

## 2.2 Eligibility

**2.2.1 Types of study.** The review will include relevant randomized trials and observational studies (including cohort and longitudinal studies) to investigate the link between gut microbiota and depression post-MI, as well as to assess the efficacy and safety of probiotic-supported treatment of depression post-MI. Excluded from the review will be cross sectional studies, case control studies, case reports, animal experiments, position papers, comments, letters, conference proceedings, policy documents, reviews, and meta-analyses.

**2.2.2 Types of participants.** Adult patients diagnosed with depression post-MI, regardless of age, gender, educational status, or ethnicity.

**2.2.3 Outcome measures.** The primary outcome measures will be differences in the profile of gut microbiota in the observational studies, and rating scale scores for depression, assessed through the use of depression scales such as HAMD, SDS, PHQ-9, and Beck Depression Inventory, at pre- and post-intervention with probiotics in the interventional trials. Secondary outcome measures will focus on the incidence of adverse cardiovascular events (MACE) in the interventional trials and the bacterial diversity including α diversity and β diversity in the observational studies.

## 2.3 Search strategy

Searches will be conducted by combining keywords or medical subject heading terms (MESH) from eligible studies in the listed databases. The same search terms will be adapted to the specific requirements of the different grammar rules. The electronic search strategy is listed in supplementary material. Eight electronic databases will be searched to identify relevant studies from inception to October 2023, including PubMed, MEDLINE, Scopus, Embase, Cochrane Clinical Trials Database, Web of Science, China National Knowledge Infrastructure (CNKI), and the China Biomedical Literature Database. "(gut microbiota OR intestinal microorganism OR gut microenvironment OR bacterial probiotic) AND (myocardial infarction OR percutaneous coronary intervention OR acute coronary syndrome) AND (depression OR depressive state OR depressive symptoms OR depressive disorder)" will be used as the keywords. Table 1 displays the detailed search strategy used in PubMed. Records will be collected from all sources on the same day.

In addition, the reference lists of included literature and relevant systematic reviews will be examined to identify qualified studies.

## 2.4 Research options

Two researchers will independently screen study titles and abstracts based on inclusion criteria. If there is any disagreement, the researchers will discuss the matter to reach a consensus. If necessity, an expert and a third researcher in the same field will be consulted.

**Table 1. Search strategy for Pubmed databases.**

| NO. | Search Terms |
| --- | --- |
| #1 | gut microbiota[title/abstract] OR intestinal flora[title/abstract] OR gut flora[title/abstract] intestinal microorganism OR enteric microorganism[title/abstract] OR gut microenvironment[title/abstract] OR gut bacteria[title/abstract] OR brain and gut microbiota axis[title/abstract],bacterial probiotic[title/abstract] OR bifidobacterium[full text] OR lactobacillus acidophilus[full text] OR bacillus filiformis[full text] |
| #2 | myocardial infarction[title/abstract] OR percutaneous coronary intervention [title/abstract] OR after percutaneous coronary intervention[title/abstract], acute coronary syndrome[title/abstract] OR ST segment elevation myocardial infarction [full text] OR non ST segment elevation myocardial infarctio [title/abstract]; unstable angina pectoris [full text] OR angina pectoris [full text] OR recurrent angina pectoris [full text] OR adverse cardiovascular events[title/abstract] |
| #3 | depression[title/abstract] OR depressive state[title/abstract] OR depressive symptoms[title/abstract] OR depressive disorder[title/abstract] OR depressive mood[title/abstract]; mild depression[title/abstract] OR moderate depression[title/abstract] OR major depression[title/abstract] OR geriatric depression[title/abstract] OR chronic depression[title/abstract] |
| #4 | #1 AND #2 AND #3 |
| #5 | randomized controlled trial[title/abstract] OR controlled clinical trial[title/abstract], randomized [title/abstract] OR placebo [full text] OR randomly[title/abstract] OR observational study[title/abstract] OR cohort study[title/abstract] OR longitudinal studies[title/abstract] |
| #6 | meta analysis [title/abstract] OR Cochrane review[title/abstract] OR systematic review[title/abstract] OR case control[title/abstract] OR animal experiment[title/abstract] OR cross-section[title/abstract] |
| #7 | #4 AND #5 NOT #6 |

## 2.5 Data extraction

The two reviewers will extract the data independently using a predefined form of data extraction. All articles will be screened and those deemed ineligible or duplicate publications by two researchers were removed, any disagreements will be resolved by a third researcher. Details will be then recorded in an Excel spreadsheet. The following data sets will be extracted: author name, year of publication, language, study object, age, study type, sample size, age, α diversity, β diversity, taxa composition at five levels, outcome measures, interventions (if exists), controls (if exists), and adverse events.

If any conflicts arise, they will be resolved through group discussion or consultation with a third reviewer. If data is found to be missing, the author will be contacted through email or other methods to obtain the data. If there is no response, we will only analyze the existing data and describe the causes and effects of this exclusion in the paper. Finally, we will use Endnote software to remove duplicates. For duplicated publications, the study with the largest number of patients will be included. Fig 1 displays the flow chart for the systematic review and meta-analysis, with additional information to be completed after the screening study is concluded.

## 2.6 Risk of bias

We will apply the Cochrane Risk of Bias Assessment Tool to evaluate the quality for the interventional trials [22]. The tool will examine random sequence generation, allocation concealment, blinding of subjects and personnel, blinding of outcome assessment, adequacy of treatment of incomplete outcome data, evidence of selective outcome reporting, and other prospective sources of bias. The quality of the observational study will be evaluated independently by the Risk of Bias Assessment Tool for Nonrandomized Studies (RoBANS) [23]. This tool will assess the selection of participants, confounding variables, measurement of exposure, blinding of outcome assessments, in-complete outcome data, and selective outcome reporting.

## 2.7 Data synthesis

Descriptive literature synthesis methods will be used to describe differences in gut microbiota composition. Meta-analyses will be performed using Rev Man V.5.3 software. For the different rating scale scores for depression, we will use standardised mean differences (SMDs) between the probiotics intervention group and control group for description. When multiple depression rating scales coexist in the same study, we will use the HAMD score to calculate the SMDs. For the bacterial diversity, it will also be described between the depression post-MI group and control group using the SMDs. And for MACE, it will be described using the relative risk (RR value). We also calculated the 95% confidence interval (CI).

Heterogeneity evaluation will include clinical heterogeneity and statistical heterogeneity. We will evaluate the presence of clinical heterogeneity among studies by comparing population characteristics (height, weight, age, gender, etc.), outcomes definitions, interventions, and study designs and methods. The $I^2$ statistic will be used to test statistical heterogeneity (low, moderate, and high $I^2$ threshold values for 25%, 50%, and 75%, respectively). Fixed-effects models will be used when both clinical heterogeneity and statistical heterogeneity are low, otherwise random-effects models will be used [24]. Sensitivity analyses can not only assess the robustness and reliability of the results, but also assess the source of heterogeneity. Excluding studies of low quality or small sample sizes, the results will be robust if heterogeneity does not change significantly. Otherwise, excluded studies may be a source of heterogeneity. Furthermore, the study will conduct subgroup analyses on the depression severity of the participants using the same psychiatric scales employed, study type and geographic region. Publication bias will be analyzed by the Egger's test and a *P*-value of < 0.05 will be considered significant.

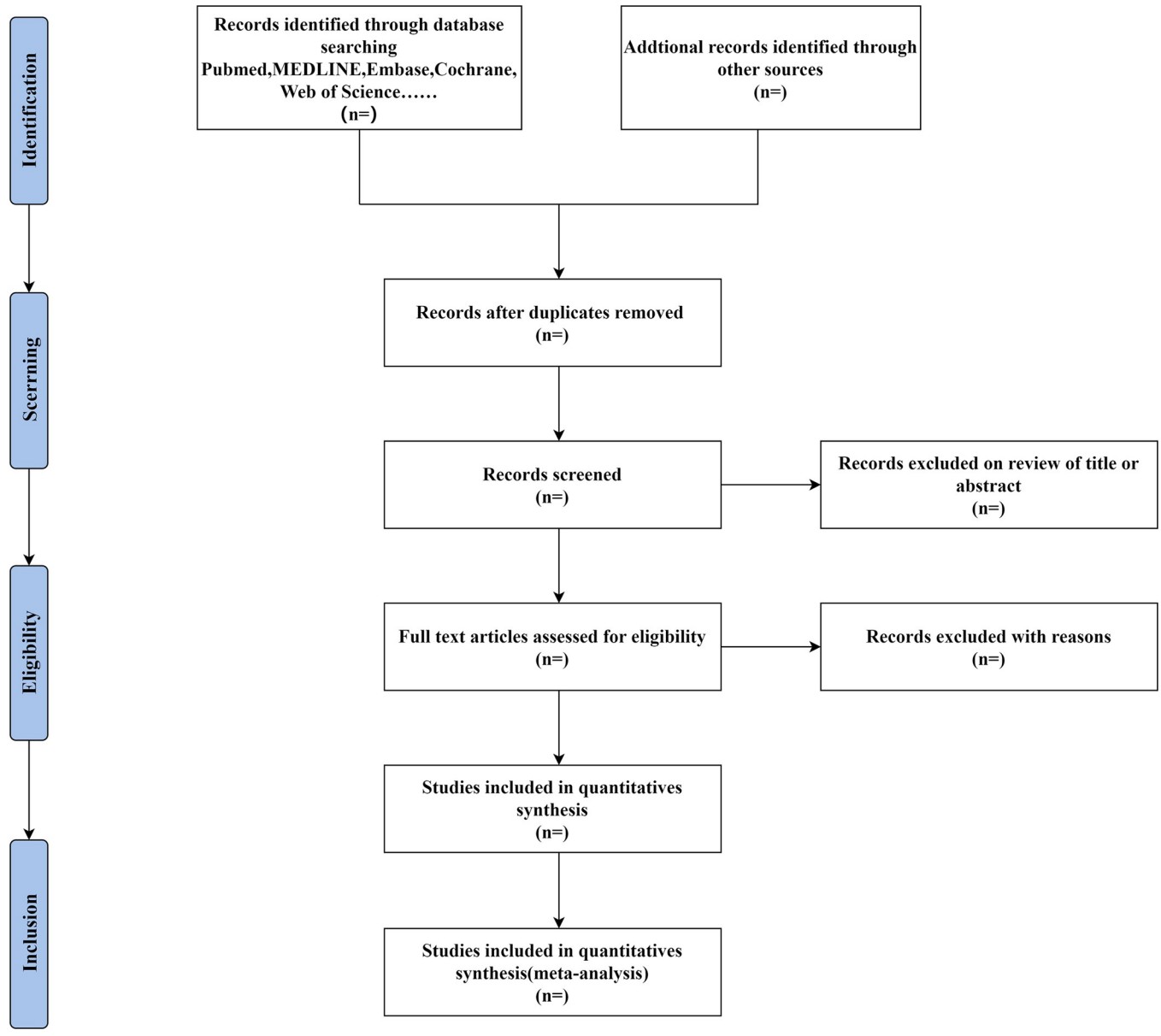

**Fig 1. Flow chart of literature retrieval, selection, and analysis.**

### 2.8 Evidence quality assessment

Grading, Assessment, Development, and Evaluation (GRADE) criteria will assess the final evidence quality of eligible studies for the outcome of interest. Each study's GRADE profile will outline six criteria: publication bias, imprecision, indirectness, effect size, inconsistency, and risk of bias. The quality of evidence will be categorized into four levels—very low, low, moderate, and high quality—to indicate the level of certainty and strength of evidence [25,26].

### 2.9 Ethics and communication

It is a secondary study that doesn't involve direct patient participation, addressing ethical concerns. The findings will be published in peer-reviewed journals.

## 3 Discussion

Despite an increase in the involvement of psychotherapy and psychopharmacological treatments over the past few decades [27–29], the prevalence and burden of anxiety and depressive disorders have persisted. In addition, there is a wide variation in response to the available treatments, which are overall are effective in less than half of the diagnosed patients [30,31]. Therefore, there is an urgent need for enhanced comprehension of the underlying pathophysiology of anxiety and depression to develop more effective treatments. Consequently, our objective is to conduct a meta-analysis to ascertain the impact of gut microbiota on depression post-MI.

The relationship between the gastrointestinal tract (GIT) and the brain has been the subject of numerous studies for years. Recently, the observation that the gut microbiota regulates the GBA sheds new light on the pathophysiological concept of disease, which has been described as a paradigm shift in neuroscience [32,33]. The microbiome may play a part in diseases characterized by psychological stress and inflammation.

Depression, a prevalent mental illness, is linked to irregularities in the structure and function of the hippocampus and prefrontal cortex in the brain. After a sudden MI, stress plays a critical role in contributing to depression. The relationship between MI and the gut microbiota has been studied extensively. Dysfunction of the HPA axis has been observed in both depressed patients and animal models of the disease [34]. Previous research suggests that the gut microbiota may play a role in the regulation of the HPA axis [35]. HPA axis dysfunction, commonly observed in depression, is correlated with elevated levels of cortisol and inflammatory molecules [36]. Consequently, this prolonged inflammation may further impact gut microbiota by the GIT. Elevated cortisol levels and inflammatory substances can compromise the blood-brain barrier (BBB) and gut permeability, promoting the translocation of gram-negative bacteria into the bloodstream and the induction of chronic inflammation in the CNS [35]. This is widely recognized as a key factor in depression post-MI. Accordingly, it appears that the role of gut microbiota in regulating emotion via the GBA could be particularly significant for depression etiology.

Recent studies have focused on the composition of gut microbiota in depressed groups, considering the role of gut microbiota in bidirectional communication between the gut and the brain. Gut microbiota abnormalities have been observed in preclinical models of rodents displaying characteristics like anxiety and depression behaviors. By giving bacterial probiotics to treat these disruptions administration, gut microbiota can return to normal and behavioral changes can be avoided [37]. When comparing the gut microbiota of depressed people to that of healthy controls, differences have been found [38]. However, these findings are at odds with the diversity of gut microbiota linked to depression. Furthermore, it remains unclear whether there are any population-based discrepancies in gut microbiota [39]. Studies reveal that the complexity of interactions between the gut microbiota and the host extends beyond the simple counting of the species of bacteria in a community [40]. It has been found that Lactobacillus rhamnosus, a commonly used probiotic, can reduce depression-related behaviors by influencing γ-aminobutyric acid in specific regions of the brain [41]. Probiotics reduced depression post-MI in rats by inhibiting the apoptotic process in the dentate gyrus and the medial amygdala of the limbic system [42], which provides additional evidence for the effects of probiotics on the GBA. A probiotic intervention study showed decrease in Corprococcus and Faecalobacterium in depressed patients compared to non-depressed controls, and an improvement in depressive symptoms. However, current clinical studies and reviews fail to sufficiently account for the impact of study quality and confounding factors. To provide a clearer understanding of their relationship, this study will conduct subgroup analysis on the participants' depression severity, study type and geographic region.

Overall, studies on gut microbiota and depression post-MI are lacking, the results of the study may provide some relevant evidence for the clinical prevention and treatment of depression post-MI with probiotics.

## Supporting information

**S1 Checklist. PRISMA-P (Preferred Reporting Items for Systematic review and Meta-Analysis Protocols) 2020 checklist: Recommended items to address in a systematic review protocol*.**
(DOC)

## Author Contributions

**Conceptualization:** Xiaojun Shi.

**Investigation:** Xiaojun Shi.

**Methodology:** Xiang Liu.

**Project administration:** Haibin Zhao.

**Resources:** Weizhe Zhao.

**Supervision:** Jiqiu Hou.

**Visualization:** Wanli Ding.

**Writing – original draft:** Xiang Liu.

**Writing – review & editing:** Xiang Liu, Xiaojun Shi.

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
