## [Decision Letter · Decision Letter 0]

19 Feb 2024

PONE-D-23-39915Association of intestinal flora with post-myocardial infarction depression: a systematic evaluation and meta-analysis protocolPLOS ONE

Dear Dr. Liu,

Thank you for submitting your manuscript to PLOS ONE. After careful consideration, we feel that it has merit but does not fully meet PLOS ONE’s publication criteria as it currently stands. Therefore, we invite you to submit a revised version of the manuscript that addresses the points raised during the review process.

We look forward to receiving your revised manuscript.

Kind regards,

António Machado

Academic Editor

PLOS ONE

Journal Requirements:

2. Thank you for stating the following financial disclosure: "the National Natural Science Foundation of China (grant number No. 82174332)."

3. Thank you for stating the following in the Acknowledgments Section of your manuscript: "This research was supported by the National Natural Science Foundation of China (grant number No. 82174332)."

Please remove any funding-related text from the manuscript and let us know how you would like to update your Funding Statement. Currently, your Funding Statement reads as follows: "the National Natural Science Foundation of China (grant number No. 82174332)."

5. Please amend the manuscript submission data (via Edit Submission) to include authors Jiqiu Hou, Weizhe Zhao, and Wanli Ding.

Additional Editor Comments:

Dear authors,

I am pleased to say that the reviewers enjoyed the manuscript very much and we are excited about the possibility of publishing your work. However, both reviewers reported several concerns and the need for major revisions in the original version of the manuscript. Please read carefully both reviewers’ reports addressing and answering all comments and suggestions.

So, I kindly invite the authors to realize a thoughtful revision of the submitted manuscript to achieve publication endorsement by the reviewers.

Thank you and best regards,

António Machado

Reviewers' comments:

Reviewer's Responses to Questions

**Comments to the Author**

1. Does the manuscript provide a valid rationale for the proposed study, with clearly identified and justified research questions?

Reviewer #1: No

Reviewer #2: Yes

2. Is the protocol technically sound and planned in a manner that will lead to a meaningful outcome and allow testing the stated hypotheses?

Reviewer #1: No

Reviewer #2: No

3. Is the methodology feasible and described in sufficient detail to allow the work to be replicable?

Reviewer #1: No

Reviewer #2: No

4. Have the authors described where all data underlying the findings will be made available when the study is complete?

Reviewer #1: No

Reviewer #2: Yes

5. Is the manuscript presented in an intelligible fashion and written in standard English?

Reviewer #1: No

Reviewer #2: No

6. Review Comments to the Author

You may also provide optional suggestions and comments to authors that they might find helpful in planning their study.

Reviewer #1: Please follow the PRISMA 2020 statement.

Since this review will include randomized trials and observational studies (metioned in 2.2.1 Type of study), which methods will be selected to analyze the data/evidence from these two types of clinical trials?

Reviewer #2: Re: Association of intestinal flora with post-myocardial infarction depression: a systematic evaluation and meta-analysis protocol

The proposed systematic review is interesting and clinically relevant. However, the current protocol lacks much needed detail. The protocol also requires English language professional editing.

Main concerns

• Study designs to be included in the SR: Observational studies such as cross sectional or case-control cannot determine temporality in a relationship. One may never know if the depression preceded or came after the MI. Since the relationship that is sought in the SR is temporal, I advise the authors to consider excluding cross sectional and case control studies.

• Search strategy is not clear: Although listed in the text, cross-sectional design is not shown in the outlined search strategy (Table 1), so it is not clear if this design will be targeted. Furthermore, case-control is excluded, whereas in the text (under 2.2.1) it is included.

• The Chinese search terms and search strategy must be provided in an appendix.

• Will duplicates be removed? From past experience, I have seen that studies published in Chinese may have their exact published duplicates although published in another language, say English. Also, the same team may numerously publish the same study. How will these be screened out?

• Data extraction (2.5), it is not clear what is meant by “efficacy measures” or “complications”. Please clarify. All study outcome measures must be listed in the data extraction sheet.

• The analysis section needs a total revision, and some sentences need a rewrite. It is not clear what the “binary effect size” relates to. Please clarify and add more detail to the statistical methods.

What is meant by the statement “When heterogeneity is high, we will first analyse and treat it”. Did you mean that factors contributing to the heterogeneity will be investigated? Also please clarify how will high heterogeneity be determined.

Change the tense of following sentence to future tense. “The source of heterogeneity was assessed by sensitivity analysis, and after excluding studies of low quality or small sample sizes, the results were robust if heterogeneity did not change significantly.”

The sentence “Otherwise, excluded studies may be a source of heterogeneity” is not clear.

The following sentence is not clear “In this study, the included studies assessed using the funnel plot have fewer than 10 publications biased, otherwise Egger regression tests will be used.”

Besides depression severity, I advise the authors to conduct sub analyses by study design and by geographic region of the study.

The planned sensitivity analysis is not clear.

How will missing data be dealt with?

• Abstract: Study measures and methods need further clarification. The main outcome measure(s) should be stated. Also please clarify by what factors will the sub-analyses be conducted. The analysis plan is not clear. What method will be employed in running forest plots? Will the authors consider random effect models? Please clarify in the Abstract. In abstract, please remove all citations.

7. PLOS authors have the option to publish the peer review history of their article (what does this mean?). If published, this will include your full peer review and any attached files.

Reviewer #1: No

Reviewer #2: No

---

## [Author Response · Author response to Decision Letter 0]

7 Apr 2024

Title: Association of intestinal flora with post-myocardial infarction depression: a systematic evaluation and meta-analysis protocol

Dear editor,

Thank you very much for your feedback on the manuscript.The comments of reviewers were extremely insightful and enabled us to greatly improve the quality of our manuscript. In the following pages are our point-by-point responses to each of the comments of the reviewers. We have revised the article according to the reviewer's suggestions, in which the revised parts are in red font, while the revision process retains the traces of revision for review process.

We have meticulously revised and polished the language of the manuscript through a professional English editing service. 

We hope that with these revisions and our responses, our manuscript is now suitable for publication in PLOS ONE.

We shall look forward to hearing from you at your earliest convenience.

Yours sincerely，

Xiang Liu

Beijing University of Traditional Chinese Medicine

2024.4.1

In relation to Journal Requirements:

When submitting your revision, I addressed these additional requirements.

1. Please ensure that your manuscript meets PLOS ONE's style requirements, including those for file naming.The PLOS ONE style templates can be found athttps://journals.plos.org/plosone/s/file?id=wjVg/PLOSOne_formatting_sample_main_body.pdfandhttps://journals.plos.org/plosone/s/file?id=ba62/PLOSOne_formatting_sample_title_authors_affiliations.pdf

Response:This manuscript conforms to your journal's style requirements.

2. Thank you for stating the following financial disclosure: "the National Natural Science Foundation of China (grant number No. 82174332)."

Please state what role the funders took in the study.  If the funders had no role, please state: "The funders had no role in study design, data collection and analysis, decision to publish, or preparation of the manuscript."If this statement is not correct you must amend it as needed.Please include this amended Role of Funder statement in your cover letter; we will change the online submission form on your behalf.

Response:The amended Role of Funder statement is as follows:The funders had no role in study design, data collection and analysis, decision to publish, or preparation of the manuscript.

3. Thank you for stating the following in the Acknowledgments Section of your manuscript: "This research was supported by the National Natural Science Foundation of China (grant number No. 82174332)."

Please remove any funding-related text from the manuscript and let us know how you would like to update your Funding Statement. Currently, your Funding Statement reads as follows: "the National Natural Science Foundation of China (grant number No. 82174332)."

Response:The funding information has been removed from the manuscript. The funding statement remains unchanged:Acknowledgments:This research was supported by the National Natural Science Foundation of China (grant number No. 82174332).

4.Please provide a complete Data Availability Statement in the submission form, ensuring you include all necessary access information or a reason for why you are unable to make your data freely accessible. If your research concerns only data provided within your submission, please write "All data are in the manuscript and/or supporting information files" as your Data Availability Statement.

Response:I modified online refer to Data Availability Statement:No datasets were generated or analysed during the current study. All relevant data from this study will be made available upon study completion.

5. Please amend the manuscript submission data (via Edit Submission) to include authors Jiqiu Hou, Weizhe Zhao, and Wanli Ding.

Response:The above author information has been amended online.

To Reviewer 1

Dear Reviewer,

Thank you for taking the time to review our manuscript and providing your constructive feedback. We sincerely appreciate your valuable comments and suggestions. In accordance with your feedback, we have revised the manuscript.The specific modifications to the content of the article are as follows.

1 Please follow the PRISMA 2020 statement.Since this review will include randomized trials and observational studies (metioned in 2.2.1 Type of study), which methods will be selected to analyze the data/evidence from these two types of clinical trials?

Response:Thank you for your valuable suggestion.Your question is of high significance.The statement in the text has been amended to "PRISMA 2020 Statement" and the cited references have been updated.With regard to the choice of the two types of research methods, according to my understanding, my explanation is as follows, and I am grateful to the teacher for criticising and correcting me.Although this review included two types of studies, Revan 5.3 was selected for meta-analysis. Among them, the protocol of this meta-analysis was formulated in terms of participant type,outcome indicator,search strategy,study selection,data extraction,risk of bias,and data synthesis. In addition, the selection of models was evaluated according to the level of heterogeneity.For example:We will examin the statistical heterogeneity using the I2 statistic. If the I2 value exceeds 50%, significant heterogeneity will be considered, and the random effects model will be employed. Otherwise, the fixed-effects model was used.At the same time, the programme mentions that subgroup analyses will be conducted according to the study type. Therefore, regardless of the type of studies included in this study, the meta-analysis will be conducted according to the steps in the protocol.The above is only a personal opinion, if there is something inappropriate, I hope the teacher do not hesitate to advise. Thank you again for your valuable advice.

To Reviewer 2

Dear Reviewer,

Thank you for your recognition of this study.Your suggestions were extremely insightful and enabled us to greatly improve the quality of our manuscript.In accordance with your suggestions, we have revised the manuscript, added much needed details,and subjected the language to rigorous editing and refinement through a professional English editing.The specific modifications to the content of the article are as follows:

Main concerns

1 Study designs to be included in the SR: Observational studies such as cross sectional or case-control cannot determine temporality in a relationship. One may never know if the depression preceded or came after the MI. Since the relationship that is sought in the SR is temporal, I advise the authors to consider excluding cross sectional and case control studies.

Response:Thanks to your valuable advice.Cross sectional and case control studies have been deleted from the text (2.2.1) as suggested.

2 Search strategy is not clear: Although listed in the text, cross-sectional design is not shown in the outlined search strategy (Table 1), so it is not clear if this design will be targeted. Furthermore, case-control is excluded, whereas in the text (under 2.2.1) it is included.

Response:Thank you for your important comments.Cross-sectional and case-control studies have been deleted in 2.2.1, and it has been checked that the reference in Table 1 is to case control[title/abstract] OR cross-section[title/abstract] in part 6# (7# has been labelled as the type of study to be excluded here).

3 The Chinese search terms and search strategy must be provided in an appendix.

Response:Thanks to your advice.Chinese search terms and search strategies have been added and uploaded in the Appendix.

4 Will duplicates be removed? From past experience, I have seen that studies published in Chinese may have their exact published duplicates although published in another language, say English. Also, the same team may numerously publish the same study. How will these be screened out?

Response: Duplicates will be removed and relevant changes were made in response to the literature selection issues mentioned in section 2.5 of the article:"All articles will be screened and those deemed ineligible or duplicate publications by two researchers were removed，any disagreements will be resolved by a third researcher. Details will be then recorded in an Excel spreadsheet. The following data sets will be extracted: author name, year of publication, language, study object,age,study type, sample size, age, outcome measures, interventions (if exists), controls (if exists),and adverse events."Thus this step can avoid duplication of included studies to some extent. And the joint participation of two researchers can avoid the omission of one person in entering the form and screening process. At the same time, Meanwhile, this study added the following statement to the section:"Finally, use Endnote software to remove duplicates.For duplicated publications, the study with the largest number of patients were included. 

5 Data extraction (2.5), it is not clear what is meant by “efficacy measures” or “complications”. Please clarify. All study outcome measures must be listed in the data extraction sheet.

Response:Thanks to your valuable advice. In the Data extraction (2.5) section of the article, the words has been corrected to "outcome measures, and adverse events" due to ambiguity caused by inaccurate words(efficacy measures or complications). Outcome measures and adverse events can be found in Table 1 Search strategy for PubMed databases.

6 The analysis section needs a total revision, and some sentences need a rewrite. It is not clear what the “binary effect size” relates to. Please clarify and add more detail to the statistical methods.

Response:Thank you for such detailed advice on this section.The analysis section has been completely revised and the relevant sentences involved have been rewritten as suggested.The specific modifications are as follows:

1)Through extensive review of the literature and careful consideration, it is found that this study does not involve binary effect size, so this part has been deleted, and we apologise for any inconvenience caused.

2)What is meant by the statement “When heterogeneity is high, we will first analyse and treat it”. Did you mean that factors contributing to the heterogeneity will be investigated? Also please clarify how will high heterogeneity be determined.

Response:We are deeply grateful for your invaluable advice.High and low heterogeneity are mainly related to model selection issues, so the heterogeneity-related statements have been modified and conditions have been added regarding the judgement of high and low heterogeneity.

3)Change the tense of following sentence to future tense. “The source of heterogeneity was assessed by sensitivity analysis, and after excluding studies of low quality or small sample sizes, the results were robust if heterogeneity did not change significantly.”

Response:Thank you very much for your suggestion.The following sentences have been changed to the future tense:After excluding studies of low quality or small sample sizes, the results will be robust if heterogeneity does not change significantly. 

4)The sentence “Otherwise, excluded studies may be a source of heterogeneity” is not clear.

Response:We are deeply grateful for your invaluable advice.The sentence mentioned above:” Otherwise, excluded studies may be a source of heterogeneity.”It is a further elaboration of the previous sentence, which aims to show that low quality or small sample size of studies may be a factor affecting heterogeneity, thus the previous sentence already mentioned that:”After excluding studies of low quality or small sample sizes, the results will be robust if heterogeneity does not change significantly”.This sentence, on the other hand, illustrates the converse of the above, aiming to show that if heterogeneity changes significantly after excluding low quality or small sample size of studies, the excluded studies are are are a source of heterogeneity. 

5)The following sentence is not clear “In this study, the included studies assessed using the funnel plot have fewer than 10 publications biased, otherwise Egger regression tests will be used.”

Response:Thank you very for this valuable suggestions.I have modified the "In this study, the included studies assessed using the funnel plot have fewer than 10 publications biased, otherwise Egger regression tests will be used. " mentioned above,and also explained the conditions for the use of funnel plot and Egger test as shown below:Given to the limited capacity of funnel plots when pooling a small number of trials, we will use funnel plots to when the included studies fewer than 10 publications, otherwise Egger’s test will be used to verify the potential publication bias.

6)Besides depression severity, I advise the authors to conduct sub analyses by study design and by geographic region of the study.

Response:Thanks to the valuable advice, the research design and the geographical region of the study have been added to the sub analyses approach.

7)The planned sensitivity analysis is not clear.

Response:Thank you for your advice.I have added the following to the sensitivity analysis section:We will also Improve stability by changing analysis models, inclusion and exclusion criteria, or by excluding a certain type of literature.At the same time, I have revised the relevant parts of the above text on sensitivity analysis and heterogeneity.

8)How will missing data be dealt with?

Response:Thank you for your question.The method of handling missing data was explained in the text (section 2.5) with the addition of relevant statements, as shown below:If data is found to be missing, the author will be contacted through email or other methods to obtain the data. If there is no response, we will only analyze the existing data and describe the causes and effects of this exclusion in the paper.

7 Abstract: Study measures and methods need further clarification. The main outcome measure(s) should be stated. Also please clarify by what factors will the sub-analyses be conducted. The analysis plan is not clear. What method will be employed in running forest plots? Will the authors consider random effect models? Please clarify in the Abstract. In abstract, please remove all citations.

Response:We are very grateful for your invaluable advice. The main outcome measures of the study, factors for which sub analyses were conducted have been added to the abstract. The use of random effect models and conditions have been described. Also, the method of data analysis has been changed to RevMan V.5.3 which in line with section 2.7 below, and Stata has been deleted. It has been stated that the random forest plots will be carried out using RevMan V.5.3. The citations in the abstract has been removed and the citation numbers of the reference has been corrected.

---

## [Decision Letter · Decision Letter 1]

24 Apr 2024

PONE-D-23-39915R1Association of intestinal flora with post-myocardial infarction depression: a systematic evaluation and meta-analysis protocolPLOS ONE

 Dear Dr. Liu,

I am pleased to inform you that both reviewers only requested minor revisions for future publication endorsement. Please carefully answer both reviewers' concerns and rectify the manuscript following their comments.

Thank you and best regards,

António Machado

Reviewers' comments:

Reviewer's Responses to Questions

**Comments to the Author**

1. Does the manuscript provide a valid rationale for the proposed study, with clearly identified and justified research questions?

Reviewer #1: No

Reviewer #2: Yes

Reviewer #3: Yes

2. Is the protocol technically sound and planned in a manner that will lead to a meaningful outcome and allow testing the stated hypotheses?

Reviewer #1: No

Reviewer #2: No

Reviewer #3: Yes

3. Is the methodology feasible and described in sufficient detail to allow the work to be replicable?

Reviewer #1: No

Reviewer #2: Yes

Reviewer #3: Yes

4. Have the authors described where all data underlying the findings will be made available when the study is complete?

Reviewer #1: No

Reviewer #2: Yes

Reviewer #3: Yes

5. Is the manuscript presented in an intelligible fashion and written in standard English?

Reviewer #1: No

Reviewer #2: No

Reviewer #3: Yes

6. Review Comments to the Author

You may also provide optional suggestions and comments to authors that they might find helpful in planning their study.

Reviewer #1: Your response did not answer my previous comment "Since this review will include randomized trials and observational studies (metioned in 2.2.1 Type of study), which methods will be selected to analyze the data/evidence from these two types of clinical trials? ".

The manuscript contains fundamental errors. The strategy of a model choice for meta-analysis on the heterogeneity result is wrong.

How to assess the quality of observational studies?

An appealing review should include a critical assessment of the relevant literature published.

Reviewer #2: I thank the authors for revising their protocol. There are still some issues that need to be addressed.

The whole section under “Data synthesis” needs major revision. It appears that sections were taken from other published sources without them being properly edited or adapted to this study. The tenses used must all be in future tense and not past tense.

Some sentences still do not make sense and must be revised. For example, the following sentence is not clear:

“Given to the limited capacity of funnel plots when pooling a small number of trials, we will use funnel plots to when the included studies fewer than 10 publications, otherwise Egger’s test will be used to verify the potential publication bias.”

Since the depression scales are continuous, it is not clear “Odds Ratios” of what will be estimated? Also, it is not clear mean difference of what measures will be estimated? I am assuming that the mean differences in depression scale scores between those who did and did not have a myocardial infarction will be estimated. This needs to be clearly stated.

The decision to run random effect models when the heterogeneity is 50% or more is a common methodological mistake that authors do. A non-significant heterogeneity test does not necessarily indicate that the studies share a common effect size. Sometimes the test of heterogeneity is underpowered and therefore can provide non-significant results even when there is heterogeneity.

The choice of a model should be based on the question of which model fits the distribution of effect sizes. The random effects model is often the more careful and credible approach that is used when meta-analysing results from different studies.

Please see Borenstein et al 2010.

• Borenstein M, Hedges LV, Higgins JP, Rothstein HR. A basic introduction to fixed-effect and random-effects models for meta-analysis. Res Synth Methods. 2010 Apr;1(2):97-111. doi: 10.1002/jrsm.12. Epub 2010 Nov 21. PMID: 26061376.

Reviewer #3: Following the recommendations of the previous reviewers, I suggest the following enhancements:

The term "flora" is outdated and should be replaced with "microbiota."

Utilize abbreviations like GBA for gut-brain axis and other necessary terms.

Trim the abstract length in accordance with the journal's guidelines.

Improve the writing; consider employing an AI tool to streamline ideas and enhance syntax.

Enhance the quality and/or resolution of the images.

Table 1 lacks substantive content; I propose integrating its description into the text.

Consider including a concise visual abstract if feasible.

Overall, it's a promising piece of work, but refinement is needed before publication.

Best regards,

7. PLOS authors have the option to publish the peer review history of their article (what does this mean?). If published, this will include your full peer review and any attached files.

Reviewer #1: No

Reviewer #2: No

Reviewer #3: **Yes: **Fausto Cabezas-Mera

---

## [Author Response · Author response to Decision Letter 1]

16 May 2024

Association of gut microbiota with depression post-myocardial infarction: a systematic evaluation and meta-analysis protocol

Dear editor/reviewer,

We would like to thank you for your efforts in reviewing our manuscript, providing many helpful suggestions and guiding our research in the future. The comments were extremely insightful and enabled us to greatly improve the quality of our manuscript.

We have studied the comments point by point, revised the manuscript accordingly. The amendments are highlighted in yellow in the revised manuscript. All authors have approved the response letter and the revised version of the manuscript. 

We hope that the revised version of the manuscript is now acceptable for publication in PLOS ONE. If you have any queries, please do not hesitate to contact me. 

Thank you again for your valuable comments and suggestions. I look forward to hearing from you soon. 

Yours sincerely，

Xiang Liu,Haibin Zhao

Beijing University of Chinese Medicine

Telephone: 18810620797

Email:460259680@qq.com;haibin999@126.com

2024-5-16

To Reviewer 1

Dear Reviewer,

We really appreciate you for your carefulness and conscientiousness. Your suggestions are really valuable and helpful for revising and improving our paper. According to your comments, we have made the following revisions on this manuscript.

The reviewer comments are laid out below in italicized font. Our response is given in normal font and amendments or supplement to the manuscript are highlighted in yellow. 

1. Your response did not answer my previous comment "Since this review will include randomized trials and observational studies (metioned in 2.2.1 Type of study), which methods will be selected to analyze the data/evidence from these two types of clinical trials? ".

Response：I am sorry that I did not understand the teacher's question before and apologize for not answering that question in my previous reply. As the data types of the two clinical trials are different, their analysis methods, and bias evaluation methods are different. I have discussed the methods used for the two trial types in detail in “2.6 Risk of bias (on page 6)” and “2.7 Data synthesis (on page 6-7)” of the manuscript respectively. At the same time, we have supplemented “2.2.3 Outcome measures (on page 4)” to explain the different measures that will be used in the two two types of clinical trials. And once again, I apologize for the inconvenience caused to your, and thank you very much for your patience in correcting me.

2. The manuscript contains fundamental errors. The strategy of a model choice for meta-analysis on the heterogeneity result is wrong.

Response: Thank you for your comments, we are very sorry for this fundamental error. According to your suggestions, we have revised the model selection strategy for the meta-analysis after careful reading of a variety of methodological articles, the revised results of which are shown in “2.7 Data synthesis (on page 7)”.

3. How to assess the quality of observational studies?

Response: Thank you for your suggestion. After consideration, we planned to assess the quality of observational studies through RoBANS. The modifications in “2.6 risk of bias (on page 6)”.

4. An appealing review should include a critical assessment of the relevant literature published.

Response: Thank you very much for your valuable comments and suggestions. Due to the fact that different studies require different treatment assessments. Therefore, we will apply the Cochrane to evaluate the quality of the interventional trials. The quality of the observational study will be evaluated independently by the Risk of Bias Assessment Tool for Nonrandomized Studies (RoBANS). This section is described in “2.6 Risk of bias (on page 6)” in the text.

Again, thank you for giving us the opportunity to strengthen our manuscript with your valuable comments and queries. We have worked hard to incorporate your feedback and hope that these revisions persuade you to accept our submission.

To Reviewer 2

Dear Reviewer,

We feel great grateful for your professional review work on our article. In accordance with your suggestions, we have made the following revisions to our manuscript. Our response is given in normal font, and modification or supplement to the manuscript are highlighted in yellow. 

1. The whole section under “2.7 Data synthesis” needs major revision. It appears that sections were taken from other published sources without them being properly edited or adapted to this study. The tenses used must all be in future tense and not past tense.

Response: Thanks for the valuable advice, I have completely revised the section and changed the tense to future tense in “2.7 Data synthesis (on page 6-7)”.

2. Some sentences still do not make sense and must be revised. For example, the following sentence is not clear:

“Given to the limited capacity of funnel plots when pooling a small number of trials, we will use funnel plots to when the included studies fewer than 10 publications, otherwise Egger’s test will be used to verify the potential publication bias.”

Response: Thanks to your reminder, considering the limitation of the funnel plot (If the number of studies is less than 10, it will be difficult to find the cause of the asymmetry), this study was planned to use Egger’s test entirety to examine potential publication bias in “the last line of 2.7 Data synthesis (on page 7)”.

3. Since the depression scales are continuous, it is not clear “Odds Ratios” of what will be estimated? Also, it is not clear mean difference of what measures will be estimated? I am assuming that the mean differences in depression scale scores between those who did and did not have a myocardial infarction will be estimated. This needs to be clearly stated.

Response: Thank you for your advice, we benefited a lot. Inspired by your comments, we consider the importance of gut microbiota in the text, previously in the “Data synthesis” did not expand the interpretation of observational studies about gut microbiota, so in addition to the depression scale, we supplemented the indicators related to gut microbiota, which can be reflected in the “2.2.3 outcome measures (on Page 4)”, for the methodological differences between different research methods are reflected in the “2.7 Data synthesis (on Page 6)”. For the description of MACE, it is more appropriate to use RR values, so we deleted the Odds Ratios. Thank you again for your positive comments and valuable suggestions to improve the quality of our manuscript. 

4. The decision to run random effect models when the heterogeneity is 50% or more is a common methodological mistake that authors do. A non-significant heterogeneity test does not necessarily indicate that the studies share a common effect size. Sometimes the test of heterogeneity is underpowered and therefore can provide non-significant results even when there is heterogeneity.

The choice of a model should be based on the question of which model fits the distribution of effect sizes. The random effects model is often the more careful and credible approach that is used when meta-analysing results from different studies.

Response: Thank you for your patience and advice. We have learnt a lot from the reference. Your suggestions are important for improving the quality of this manuscript. I have made modifications for model selection in “2.7 Data synthesis (on Page 7)”.

We deeply appreciate your valuable comments and suggestions, and look forward to hearing from you regarding our submission. We would be glad to respond to any further questions and comments that you may have.

To Reviewer 3

Dear Reviewer,

We feel great thanks for your professional review work on our article. According to your comments, we have made extensive modifications to our manuscript point to point. In accordance with your suggestions, we have made the following revisions to our manuscript. Our response is given in normal font and changes or additions to the manuscript are highlighted in yellow. In addition, in order to comply with the word count requirements in the guidelines, we streamlined the text, and the deleted traces was retained for viewing.

1. Following the recommendations of the previous reviewers, I suggest the following enhancements: The term "flora" is outdated and should be replaced with "microbiota."

Response: We sincerely thank the reviewer for careful reading. As suggested by the reviewer, we have corrected “flora” into “microbiota”. The modifications are highlighted in yellow.

2. Utilize abbreviations like GBA for gut-brain axis and other necessary terms.

Response: Thanks for your careful checks. Abbreviations have been made throughout the text, such as GBA, MI, CAD. 

3. Trim the abstract length in accordance with the journal's guidelines.

Response: Abstract lengths have been trimmed according to journal guidelines. Our response is given in normal font and changes or additions to the manuscript are highlighted in yellow. In addition, in order to control the word count, we streamlined the text, and the deleted traces retained for reviewing.

4. Improve the writing; consider employing an AI tool to streamline ideas and enhance syntax.

Response: Thanks for your suggestion. With tools such as AI, we tried our best to modify and improve the manuscript. The modifications are highlighted in yellow.

5. Enhance the quality and/or resolution of the images.

Response: Thank you for your valuable suggestion. We have improved the image quality. I have uploaded the original image with an attachment in system to prevent the image from affecting the resolution after inserting into the manuscript.

6. Table 1 lacks substantive content; I propose integrating its description into the text.

Response: We sincerely appreciate the valuable comments. According to the suggestion, we have incorporated the search strategy for the keywords mentioned in Table 1 into the text. Due to word limitations, it was difficult to incorporate all the content in the table into the text, so the keywords in the text have been streamlined. At the same time, in order not to take up more words, we have replaced table 1 with an image to be inserted into the text, as shown in “2.3 Search Strategy (on Page 4-5)”.

7. Consider including a concise visual abstract if feasible.

Response: We think this is an excellent suggestion. However, given the partial duplication between the visual summary and Figure 1, we did not choose to use the visual summary for further presentation. We have simplified the abstract to make it clearer. The modifications are visible at the “Abstract (on Page 1-2)”.

8. Overall, it's a promising piece of work, but refinement is needed before publication.

Response: Thank you again for your constructive comments and suggestions on our manuscript. We hope you will find our revised manuscript acceptable for publication.

---

## [Decision Letter · Decision Letter 2]

30 May 2024

Association of gut microbiota with depression post-myocardial infarction: a systematic evaluation and meta-analysis protocol

PONE-D-23-39915R2

Dear authors,

I am pleased to inform you that both reviewers enjoyed the manuscript very much and endorsed the revised manuscript for publication.

Thank you for choosing Plos ONE journal to publish your study.

Best regards,

António Machado

Reviewers' comments:

Reviewer's Responses to Questions

**Comments to the Author**

1. Does the manuscript provide a valid rationale for the proposed study, with clearly identified and justified research questions?

Reviewer #2: Yes

Reviewer #3: Yes

2. Is the protocol technically sound and planned in a manner that will lead to a meaningful outcome and allow testing the stated hypotheses?

Reviewer #2: Yes

Reviewer #3: Yes

3. Is the methodology feasible and described in sufficient detail to allow the work to be replicable?

Reviewer #2: Yes

Reviewer #3: Yes

4. Have the authors described where all data underlying the findings will be made available when the study is complete?

Reviewer #2: Yes

Reviewer #3: Yes

5. Is the manuscript presented in an intelligible fashion and written in standard English?

Reviewer #2: Yes

Reviewer #3: Yes

6. Review Comments to the Author

You may also provide optional suggestions and comments to authors that they might find helpful in planning their study.

Reviewer #2: There are still some grammatical mistakes, and some sentences still need to be revised. I advise the authors to have the manuscript proofread by an English-speaking editor.

Reviewer #3: All comments from the previous review have been addressed correctly.

I have no new comments on the work.

7. PLOS authors have the option to publish the peer review history of their article (what does this mean?). If published, this will include your full peer review and any attached files.

Reviewer #2: No

Reviewer #3: **Yes: **Fausto Cabezas-Mera

---

## [Editor Report · Acceptance letter]

2 Aug 2024

PONE-D-23-39915R2 

PLOS ONE

Dear Dr. Liu, 

I'm pleased to inform you that your manuscript has been deemed suitable for publication in PLOS ONE. Congratulations! Your manuscript is now being handed over to our production team.

Kind regards, 

on behalf of

Dr. António Machado 

Academic Editor

PLOS ONE